# Design of Two New Sulfur Derivatives of Perezone: In Silico Study Simulation Targeting PARP-1 and In Vitro Study Validation Using Cancer Cell Lines

**DOI:** 10.3390/ijms25020868

**Published:** 2024-01-10

**Authors:** Alejandro Rubiales-Martínez, Joel Martínez, Elvia Mera-Jiménez, Javier Pérez-Flores, Guillermo Téllez-Isaías, René Miranda Ruvalcaba, Maricarmen Hernández-Rodríguez, Teresa Mancilla Percino, Martha Edith Macías Pérez, María Inés Nicolás-Vázquez

**Affiliations:** 1Departamento de Ciencias Químicas, Facultad de Estudios Superiores Cuautitlán Campo 1, Universidad Nacional Autónoma de México, Avenida 1o de Mayo s/n, Colonia Santa María las Torres, Cuautitlán Izcalli 54740, Mexico; alex91hirdck@gmail.com (A.R.-M.); atlanta126@gmail.com (J.M.); mirruv@yahoo.com.mx (R.M.R.); 2Laboratorio de Cultivo Celular, Escuela Superior de Medicina, Instituto Politécnico Nacional, Plan de San Luis y Díaz Mirón s/n, Ciudad de México 11340, Mexico; elviamj@gmail.com (E.M.-J.); dra.hernandez.ipn@gmail.com (M.H.-R.); 3Laboratorio de Espectrometría de Masas, Instituto de Química, Universidad Nacional Autónoma de México, Circuito Exterior s/n, Ciudad Universitaria, Alcaldía Coyoacán, Ciudad de México 04510, Mexico; jape@unam.mx; 4Department of Poultry Science, University of Arkansas, Fayetteville, AR 72701, USA; gtellez@uark.edu; 5Chemistry Department, Centro de Investigación y de Estudios Avanzados del Instituto Politécnico Nacional, Alcaldía Gustavo A. Madero, Ciudad de México 07000, Mexico

**Keywords:** perezone, sulfur derivates, natural products, cytotoxic activity, DFT, PARP-1

## Abstract

Poly-ADP-Ribose Polymerase (PARP-1) is an overexpressed enzyme in several carcinomas; consequently, the design of PARP-1 inhibitors has acquired special attention. Hence, in the present study, three compounds (**8**–**10**) were produced through a Michael addition protocol, using phenylmethanethiol, 5-fluoro-2-mercaptobenzyl alcohol, and 4-mercaptophenylacetic acid, respectively, as nucleophiles and perezone as the substrate, expecting them to be convenient candidates that inhibit PARP-1. It is convenient to note that in the first stage of the whole study, the molecular dynamics (MD) simulations and the quantum chemistry studies of four secondary metabolites, i.e., perezone (**1**), perezone angelate (**2**), hydroxyperezone (**3**), and hydroxyperezone monoangelate (**4**), were performed, to investigate their interactions in the active site of PARP-1. Complementarily, a docking study of a set of eleven sulfur derivatives of perezone (**5**–**15**) was projected to explore novel compounds, with remarkable affinity to PARP-1. The molecules **8**–**10** provided the most adequate results; therefore, they were evaluated in vitro to determine their activity towards PARP-1, with **9** having the best IC_50_ (0.317 µM) value. Additionally, theoretical calculations were carried out using the density functional theory (DFT) with the hybrid method B3LYP with a set of base functions 6-311++G(d,p), and the reactivity properties were compared between the natural derivatives of perezone and the three synthesized compounds, and the obtained results exhibited that **9** has the best properties to bind with PARP-1. Finally, it is important to mention that **9** displays significant inhibitory activity against MDA-MB-231 and MCF-7 cells, i.e., 145.01 and 83.17 µM, respectively.

## 1. Introduction

Cancer is the principal cause of death worldwide in countries of all income levels [1]. Cells turn cancerous after the accumulation of several mutations in their genes that control cell division and survival [2]. A wide variety of proteins have been implicated in the establishment and progression of cancer. Recently, particular attention has been paid to Poly-ADP-Ribose Polymerase 1 (PARP-1), due to its implication in the development and progress of cancer [3]. PARP-1 is a nuclear enzyme, which interacts with DNA and regulate several cell functions, such as DNA repair, recombination, proliferation, and genomic stability [4]. It has been well documented that PARP-1 is overexpressed in various carcinomas. Hence, the development of PARP-1 inhibitors has acquired importance since it has been demonstrated that this kind of molecules could induce cell death in cancer cells [5]. PARP-1 is an enzyme that facilitates the post-translational modification of proteins in response to DNA damage and genotoxic stress [6]. PARP-1 is formed by three domains: (1) a DNA-binding domain, (2) an auto-modification domain (AMD) with a role in the interaction between PARP-1 and its partner proteins, and (3) a catalytic domain [7]. Clinical PARP-1 inhibitors bind to a catalytic pocket, where they directly interfere with ADP-ribosylation [8]. The active site of PARP-1 is formed by a sequence of 50 conserved amino acids, and this region is identical for all PARP-1 sequences of vertebrates [6]. The catalytic pocket is formed by a conserved catalytic triad consisting of Hys201, Tyr235, and Glu327, which are required for the catalytic activity of PARP-1 [4]. Thus, classically, PARP-1 inhibitors are designed to target these amino acid residues.

Perezone represents a phyto compound occurring in the roots of plant specimens of the genus *Acourtia* [9]. It has been demonstrated that perezone (**1**) and natural compounds chemically related to it, such as perezone angelate (**2**), hydroxyperezone (**3**), and hydroxyperezone monoangelate (**4**) (Figure 1), induce apoptosis in several cancer cell lines in a range of micromolar concentrations [10,11,12,13,14]. Through in vitro and in silico studies, it has also been demonstrated that two mechanisms explain the pro-apoptotic activity of perezone: PARP-1 inhibition and the induction of an oxidative stress state [12]. To develop novel PARP-1 inhibitors, in the present study, molecular dynamics (MD) and quantum chemistry studies of **1**–**4** in the active site of PARP-1 were performed to investigate the interactions that drive the protein–ligand recognition, and this knowledge is crucial to design novel PARP-1 inhibitors. Then, a set of perezone derivatives were designed, highlighting that only three compounds, with the highest affinity to PARP-1 based on docking studies, were selected. Finally, the PARP-1 inhibition and its effects on several cancer cell lines were evaluated for selected compounds.

## 2. Results and Discussion

### 2.1. MD Simulations

MD allows the monitoring of binding interactions for a specific ligand with a particular protein during a defined time. As it has been previously demonstrated [13,14], **1**–**4** display IC_50_ values of 181.5 µM, 5.25 µM, 320.3 µM, and 28.7 µM, respectively. The analysis of IC_50_ of selected compounds allows the observation that the hydroxyl substituent in **3** diminishes the potency as PARP-1 inhibitor in comparison with **1**. However, the addition of an α,β-unsaturated ester moiety as an angelate, **2** and **4**, increases the PARP-1 inhibitory activity in comparison with **1**, highlighting the higher PARP-1 inhibitory activity of **2**. Consequently, the four protein–ligand complexes were subjected to 50 ns of MD simulations to investigate their interactions in the active site of PARP-1 at the molecular level.

Figure 2a shows the RMSD values of PARP-1 in the absence and the presence of compounds **1**–**4**. The determination of the RMSD of the protein alone and its complex with the studied compounds allows the quantitative evaluation of the changes in the protein structure, throughout the simulation time [15], thus determining if the simulation reaches the convergence [16]. As can be seen in Figure 2a (green), the RMSD of PARP-1 fluctuated from 2.9 to 4.1 Å, with a mean value of 3.3 Å, reaching an equilibrium at 10 ns. In comparison, the RMSD of the protein in complex with **1** fluctuated from 2.7 to 4.1 Å with a mean value of 3.1 Å, reaching an equilibrium at 10 ns (Figure 2a, pink), while the RMSD of the protein in complex with **3** ranged from 2.7 to 3.5 Å, with a mean value of 2.9 Å, reaching an equilibrium at 8 ns (Figure 2a, red). In contrast, the RMSD of the protein in complex with **2** fluctuated from 2.9 to 3.7 Å, without reaching an equilibrium (Figure 2a, gray), while the RMSD of the protein in complex with **4** ranged from 3.2 to 5.0 Å, without reaching an equilibrium (Figure 2a, blue). RMSD values showed that the presence of compounds with the highest PARP-1 inhibitory activity (**2** and **4**) prevent PARP-1 from reaching the equilibrium, and this is evidenced by higher RMSD values.

Based on the RMSF results, it is possible to demonstrate that the studied compounds induce great conformational changes in PARP-1, mainly at the N-terminal region (Figure 2b), and this is evidenced by higher RMSF values. Interestingly, residues from the catalytic triad (Hys201, Tyr235, and Glu327) show low RMSF values (Figure 2b, red arrows).

Figure 3 shows the binding mode of the studied compounds to the catalytic site of PARP-1, showing that larger compounds **2** and **4** fit deep in the catalytic site in contrast to compounds **1** and **3**. Additionally, the binding mode of **2** is maintained throughout the 50 ns of the simulation, thus explaining its good PARP-1 inhibitory activity.

As shown in Appendix A, the molecule **2** interacts with Hys201 and Tyr235 located at the catalytic site, and interestingly, these interactions are maintained during the entire simulation, supporting the higher PARP-1 inhibitory activity that was previously mentioned.

The analysis of the 50 ns snapshot of MD simulations shows that **1** exhibits hydrophobic interactions with the following amino acid residues: Ala167, Asn347, Phe348, and Asn347. In addition, both a hydrogen bond interaction with Asn347 (2.7 Å) and one π-π type interaction, between the quinonic ring of perezone and the amino acid residue ring of Tyr168, are observed. Compound **2** displays four interactions of hydrogen bonds with the following amino acid residues: Asn207 (1.7 Å), Gly202 (2.6 Å), Gly233 (2.3 Å), and Arg217 (2.2 Å). Also, it presents hydrophobic interactions with the amino acid residues Ser203, Ile234, Tyr246, Gly233, Tyr235, Hys201, and Asn106. Compound **3** has two interactions of hydrogen bonds, with the following amino acid residues: Gly227 (2.4 Å) and Asp105 (1.8 Å); in addition, the quinonic ring of hydroxyperezone displays π-π type interactions with the amino acid residue ring of Tyr228. Also, it has hydrophobic interactions with the following amino acid residues: Tyr235, Ser203, Asp105, Hys201, and Gly202. Finally, compound **4** shows four hydrogen bond interactions with the following amino acid residues: Ile234 (1.7 Å), Ser203 (2.7 Å), Tyr235 (1.9 Å), and Gly233 (1.9 Å). Additionally, hydrophobic interactions are observed with residues Ile234, Lys232, Tyr235, Ser203, Hys201, and Tyr246, which also present one π-π interaction with the ring of Tyr246.

### 2.2. Quantum Chemistry

#### 2.2.1. Molecular Electrostatic Potential (MPE)

The MEP shows the electron density of a molecule, and it is used to identify the sites of positive and negative electrostatic potentials for nucleophilic and electrophilic attacks. It is also a powerful tool to identify biological recognition processes and hydrogen bonding interactions [17]. Figure 4 shows the MEP of the molecules **1**–**4**, and red areas represent the regions of high electron density, and this is principally visualized in both carbonyl groups of the quinonic ring of all molecules, and for **3** and **4**, the density is also displayed in the carbonyl of the ester moiety. The blue areas represent the regions of low electron density; this can be seen in the hydrogen atoms of the hydroxyl groups. MEP is indicative of sites, both negative (red) or positive (blue), on molecules (ligands) where they could favor the formation of hydrophobic interactions, π-π type interactions, or hydrogen bonds towards a protein. Thus, due to a high electron density and a higher MEP value of **2**, these features contributed to the development of interactions with amino acids, highlighting the four hydrogen bonds described above and resulting in good PARP-1 inhibitory activity. In contrast, **1**, which shows a low electron density zone, displays poor interactions with amino acids, highlighting the four hydrophobic interactions and diminishing the PARP-1 inhibitory activity.

#### 2.2.2. Charge

Charge analysis is a powerful tool to investigate the intramolecular charge transfer that occurs between atoms in a molecule, to identify atoms or groups of atoms that play the role of electron donor or acceptor [18]. In the electrostatic theory, atoms with more negative charges donate electrons to atoms with higher positive charges [19]. A charge delocalization is observed on the carbon atoms of the quinone ring; therefore, the oxygen atoms O1, O2, and O3 (Figure 1) show the most negative values for the natural products derived from perezone, Appendix A. The hydrogen atom of the hydroxyl group of **1** is the most deficient in charge (0.508 e^−^). Additionally, the oxygens of the carbonyl groups of the quinone ring and the hydroxyl group display higher charges (−0.494 to −0.654 e^−^). Consequently, the hydrogen atom of the hydroxyl group can form hydrogen bonds, confirming the interaction by hydrogen bond with the amino acid residues of PARP-1. In addition, oxygen from the carbonyl of the ester group of compounds **2** and **4** interacts with amino acid residues Arg217 and Tyr235 to form hydrogen bonds, thereby confirming the hydrogen bond interaction with the amino acid residues of PARP-1 and supporting the good PARP-1 inhibitory activity.

Thus, taking into account the molecular dynamics, molecular electrostatic potentials, and charge results, in the next section, the design of a set of sulfur derivatives of perezone is described, to improve the PARP-1 inhibitory activity and to compare the activity with the natural derivatives of perezone (**2**–**4**).

### 2.3. Design of Compounds

According to the obtained results, it was possible to make several observations that allow the explanation of the affinity of the studied compounds towards the PARP-1 protein. In the first instance, the presence of a large and deep cavity in the protein allowed compounds to fit deeply; this fact is promoted by the establishment of non-bonding interactions such as π-π interaction, hydrogen bonds, and hydrophobic contacts. The presence of the angelate moiety in **2** and **4** contributes to an increase in the affinity to PARP-1. Thus, the angelate substituent establishes interactions with a non-polar aliphatic moiety of amino acid residues existent in the binding cavity, indicating that the size of the protein cavity is important for compound selectivity as has been reported for other enzymes [13,14]. Also, compounds **2** and **4** establish additional hydrogen bonding, π-π type, and hydrophobic interactions with Tyr235, and Hys201, which could be correlated with their high potency in comparison to **1** and **3**. It is also worth noting that these interactions are important because they favor an affinity between the protein and the ligand [13,14]. The analysis of the computational studies allows us to propose that hydrophobic moiety substituents could allow the orientation of the quinonic ring of perezone with Hys201 to generate a better π-π interaction, thus developing novel PARP-1 inhibitors, and aliphatic and aromatic sulfur substituents were selected to design a family of perezone derivatives, Figure 5.

### 2.4. Docking Studies

From this study, it is possible to predict and computationally calculate the most favorable position of the interaction of the sulfur perezone (ligand) molecules with the enzyme PARP-1 from their three-dimensional representations [20,21]. Figure 6 shows the free binding energies (ΔG) between the PARP-1 protein with each of the derivatives of sulfur perezone (−11.78 to −5.71 kcal/mol). This parameter represents the energy that is released when intermolecular interactions between a protein and a ligand occur. In addition, it is observed that the compounds (**8**–**10**) show a lower free binding energy, and consequently, they display a higher affinity in the interaction (ligand–protein). In contrast, compounds **5**–**7** and **11**–**15** display higher free binding energies; thus, these compounds exhibit a lower interaction with the protein. Therefore, **8**–**10** were selected to continue with the next stage of study.

The ligands that have a greater stability of interaction with the protein are those that have a lower energy requirement. In one sense, the sulfur perezone derivative **8** displays the best ΔG value, i.e., −11.78 kcal/mol, followed by **9** and **10**, with −9.79 and −9.53 kcal/mol, respectively. Appendix A shows the interaction among **8**–**10** sulfur perezone molecules and the amino acid residues of PARP-1. It is important to highlight that these molecules interact with two amino acid residues, Hys201 and Tyr235, as was previously described for natural derivatives (**2**–**4**).

Figure 7 shows the interactions between the ligand **8** and the amino acid residues of PARP-1. Three hydrogen bonds are observed with Trp200 (2.3 Å), Tyr235 (2.2 Å), and Arg217 (2.0 Å). Additionally, two π-π type interactions are observed with Hys201 and Tyr235. Hydrophobic interactions between the side chain of the ligand and the following amino acid residues are also present: Phe236, Gly202, Tyr235, Ser203, Arg217, Trp200, and Ala237.

Figure 8 displays, for molecule **9**, three hydrogen bond interactions with the following amino acid residues: Tyr228 (2.1 Å), Hys201 (1.8 Å), and Arg217 (2.0 Å). Furthermore, it depicts two π-π interactions with the following amino acid residues: Tyr235 and Tyr228. Finally, it also exhibits hydrophobic interactions of the ligand side chain with the following amino acid residues: Tyr235, Phe236, Hys201, Ser243, Ile234, and Tyr246.

Compound **10**, in Figure 9, shows three hydrogen bonding interactions with the following amino acid residues: Gly202 (2.3 Å), Gly233 (1.9 Å), and Lys242 (2.0 Å); moreover, one π-π interaction with Tyr246 is detected. Finally, five hydrophobic interactions are observed between the side chain of the ligand with the following amino acid residues: Hys201, Tyr235, Ile234, Gly233, and Ser203.

According to the structures of the sulfur derivatives of perezone, **8**–**10**, all of them exhibit a lower free binding energy, and it is important to highlight that their main feature is that the three compounds bear an aromatic ring; consequently, the π-π interactions could be improved, see above. Moreover, for **8** and **9**, there are hydrogen bond interaction types that do not exist in **2**–**4**. In general, it is important to remark that hydrophobic alkyl-π interactions are generated between the double bond of the side chain of perezone and the alkyl moiety of the amino acid residue. Hydrogen bonds are formed, for **8** and **9,** between the hydroxyl and carbonyl groups and the hydrogen of the amino groups of the amino acid residues, and for **10**, the hydrogen interactions are developed between the side hydrogen of the aliphatic chain of perezone and the carbonyl group of Gly202 and between the carbonyl groups and the hydrogen of the amine group and the alkyl chain of the amino acid residue.

Additionally, the amino acid residue Tyr235 presents three types of interactions with **8**: hydrogen bond, π-π, and hydrophobic. The Hys201 amino acid residue displays two types of interactions with **9**: hydrogen bond and hydrophobic. Compound **10** only displays hydrophobic interactions with both amino acids. Thus, it is logical to think that these compounds could offer a better inhibitory activity. In comparison with Olaparib, the compounds **8**–**10** interact with two amino acid residues from the catalytic pocket, Tyr235 and Hys201, and Olaparib only interacts with Tyr235; however, it is important to highlight that this residue is bonded by a peptidic bond with some amino acids (Ile234, Phe236, Gly233, and Ala237), favoring all types of interactions (hydrogen bond, π-π, and hydrophobic) with almost all Olaparib structures, and probably this fact favors their lower inhibitory concentration, see below.

### 2.5. Synthesis and Spectroscopic Attribution of ***9*** and ***10***

To offer a better understanding of both the synthesis and the spectroscopical characterization of the new molecules **9** and **10**, Figure 1 is provided.

The corresponding experimental data of **8** has been previously described by our research group [22]. Consequently, the structural attribution for the new molecules **9** and **10** is presented. The compound **9** is obtained as a red oil with 45% yield of pure compound. The infrared spectrophotometric data show a broad medium band centered at 3303 cm^−1^, which is assigned to OH groups, a weak band at 2050 cm^−1^ is assigned to the C=C of the aromatic moiety, two medium bands are observed at 1724 and 1600 cm^−1^ and are assigned to C=O group and C=C system, and at 739 cm^−1^, a medium band is assigned to the C-S bond. The corresponding ^1^H NMR shows a double doublet signal at 7.38–7.33 ppm and is assigned to H21, and this signal displays *ortho* coupling with H20, with a *J* = 6 Hz. At 7.28 ppm, a singlet signal was observed for the OH groups. In the range of 7.25–7.24 ppm, a doublet signal is observed for H18, showing a *meta* coupling with H20 (*J* = 3 Hz), and at 6.96–6.89 ppm, a triplet of doublet signal displaying a *meta* coupling with H18 (*J* = 3 Hz) is assigned to H20. A triplet signal at 5.28 ppm is assigned to H12. Singlet signals for H22, H7, H14, and H15 are located at 4.74, 2.19, 1.64, and 1.50 ppm, respectively. Two multiplets are assigned for H8 and H10–H11 at 2.95 and 1.80–1.64 ppm, respectively. A doublet signal is observed at 1.09 ppm and is assigned to H9. In the ^13^C NMR spectrum, the main signals that agree with the structure of **9** are assigned to the carbonyl groups at 165.4 and 162.1 ppm and the carbonyl groups of the quinonic ring. Also, the signal for aromatic carbons is from 145.7 to 115.0 ppm. The signal at 62.6 ppm is assigned to C22 for methylene carbon. Additionally, the carbon–sulfur base, C6, is observed at 145.8 ppm. Finally, the corresponding mass spectrum (EIMS) shows the ion fragment *m*/*z* 404 related to the molecular ion, and this ion is validated by HRMS-DART^+^ data, offering an elemental composition of C_22_H_26_F_1_O_4_S_1_, in agreement with a fragment ion [M + 1]^+^, and this fragment is correlated with an exact value of 405.15313 Da, a precise value of 405.15358 Da, and an error of −1.12 ppm, and complementarily, the provided unsaturation data, 10.5, is in agreement with the structure.

Compound **10** is produced as a red with a 65% yield of pure compound. The infrared spectrophotometric data show a broad medium band centered at 3372 cm^−1^, which is assigned to the CO_2_H group, a weak band at 2110 cm^−1^ that is assigned to the C=C of the aromatic moiety, two intense bands are observed at 1707 and 1637 cm^−1^ and are assigned to the C=O group and the C=C system, and at 738 cm^−1^, a band is assigned to the C-S bond. The corresponding ^1^H NMR spectrum displays a broad singlet signal at 13.83 ppm, which is unequivocally assigned to the proton of carboxylic acid, and in a range of 7.22–7.19 and 7.14–7.12 ppm, two doublet signals are presented and assigned to aromatic protons with the coupling data of *J* = 6 Hz, representative for *ortho* coupling. At 7.06 ppm, the singlet signal for the OH group is observed. Triplet, singlet, and multiplet signals are presented at 4.95, 3.54, and 2.94 ppm, and are assigned to H12, H22, and H8, respectively. The singlets for H7, H14, and H15 are localized at 2.02, 1.56, and 1.44 ppm, respectively. The signal for H10 and H11 is observed as a multiplet at 1.83–1.63 ppm; finally, a doublet signal for H9 is located at 1.07 ppm. In the ^13^C NMR spectrum, the main signals that agree with the structure of **10** are assigned to the carbonyl groups at 182.5, 181.7, and 176.7 ppm and are assigned to the carbonyl groups of the quinonic ring and the carboxylic moiety, respectively. Also, the signal at 40.5 ppm is assigned to C22, a methylene carbon type. Additionally, the signal at 150.9 ppm is observed for C6, a carbon–sulfur base. Finally, the EIMS shows the ion fragment *m*/*z* 414 in agreement with the molecular ion, and this ion is unequivocally corroborated by the HRMS-DART^+^ study, providing an elemental composition of C_23_H_27_O_5_S_1_, in agreement with a fragment ion [M + 1]^+^, and this fragment is correlated with an exact value of 415.15669 Da and a precise value 415.15792 Da (error: −2.96 ppm); complementarily, the provided unsaturation data, 11.5, is in agreement with the structure.

### 2.6. In Vitro Cytotoxic Assay

In terms of the cytotoxic activity, compounds **8** and **9** display an inhibitory activity against MCF-7 and Hela cancer cell lines, Table 1. The inhibitory concentration 50 (IC_50_) of compound **8** for Hela cells is 10.18 µM, which in comparison with **9** and **10** offers a better response. But compound **9**, showing an IC_50_ = 83.17 µM against MCF-7 cells, when compared with **8** and **10**, displays a good performance. In addition, the concentration–response charts are provided in Appendix A.

### 2.7. PARP-1 Inhibition Assay

It is well known that the major inhibitor of the PARP-1 enzyme is Olaparib, with an IC_50_ of 5.5 nM (±0.5 μM) [13]. In one sense, **8**–**10** exhibit excellent PARP-1 inhibitory activity with IC_50_ of 5.205, 0.317, and 0.320 μM, respectively, Table 2. Thus, they show a higher potency than **1** (181.5 µM) and their natural perezone derivatives **2**–**4** (5.25, 320.3, and 28.7 µM) [13,14].

With the aforementioned commentaries, it is important to mention that with the addition of an aromatic substituent to **1**, in particular with groups that can form hydrogen bonds or π interactions, the concentration needed to inhibit PARP-1 could diminish.

### 2.8. Quantum Properties and HOMO–LUMO Surface of ***2***–***4*** and ***8***–***10***

Table 3 shows the results obtained from the energies of the frontier orbitals, which are mainly involved in the chemical stability of the compound. The highest occupied molecular orbital (HOMO) represents the region that donates electrons, and the lowest unoccupied molecular orbital (LUMO) represents the region that accept electrons [23,24]. In addition, the bandgap energy (Gap) is calculated. Band gap energy is an indicator of chemical reactivity, hardness, and biological activity of molecular systems. Thus, if the bandgap energy is relatively low, it implies that a molecule can be easily polarized and attains a greater biological activity [25]. For a higher band gap energy, the molecule shows a high hardness. According to Table 3, The Gap energies are located between 2.554 and 2.853 eV. Compound **9** has the lowest band gap energy of 2.554 eV; consequently, this molecule is the most reactive of all molecules, according to the order from the highest to the lowest Gap energies of the studied molecules: **8** > **2** > **3** > **10** > **4** > **1** > **9**. In addition, according to the Principle of Maximum Hardness, it is established that the more reactive systems present low hardness values, while the less reactive systems show high hardness values. The most reactive molecules show a hardness ranging from 2.80 to 2.85 (**8**–**10**). According to the Gap and the hardness data of the seven molecules, they could show chemical and possibly biological activity since they can be reactive to change [25]. In general, for all studied compounds **1**–**4** and **8**–**10**, the HOMO orbital is located on the aliphatic side chain, in addition to the C2 and C3 atoms of the quinonic ring (Figure 10). Moreover, for **9** and **10**, the HOMO is present in the quinonic ring and also the sulfur atom. In contrast, the LUMO orbital of all molecules is distributed in the quinone system (Figure 10). These results confirm that **9** is the molecule with a better inhibitory activity towards the PARP-1 enzyme.

### 2.9. Reactivity Properties of ***2***–***4*** and ***8***–***10***

Table 3 shows the energies of chemical systems in a neutral state and with negative and positive ions. From these data, the reactivity parameters are determined in the DFT scheme. In addition, the electronic affinity (EA) values were calculated for the isolated molecules in the gas phase. The method used for calculating the electron affinity utilizes vertical electron affinities, which are obtained as the difference between the energies of the appropriate neutral form (E _Neutral_) and the anion form (E _Anion_) at optimized neutral geometries. EA is positive for all molecules explored in this study (1.79–2.14 eV). Therefore, the molecules release energy when an electron is added to a neutral molecule. It is known that the ionization potential (IP) is associated with the antioxidant capacity [26] of some compounds since this quantity is directly related to one of the mechanisms that drives the oxidation process, i.e., electron transfer. Thus, as seen in Table 3, all compounds exhibit values for this property, in the range of 7.60–8.13 kcal/mol. Due to the lowest IP of **8**, it is the easiest to oxidize. The average value of the HOMO and LUMO energies is related to the electronegativity (χ) defined by Mulliken [27] with χ = (IP + EA)/2. Electronegativity is associated with the free movement of electrons in compounds, i.e., electronegativity shows how the electrons will flow from high electronic density regions in a molecule to other sites of lower electronic density. Low global electronegativity values or high chemical potential values mean that an electron is delocalized on the molecule and the molecule can easily supply electrons to interact intra- or inter-molecularly [28]. Chemical reactivity increases as electronegativity increases [25]. The range of electronegativity is from 4.75 to 4.98. This means that **9** has a high tendency to attract electrons during interaction with another chemical compound. The global electrophilicity index, ω, measures the stabilization in energy when the molecule acquires an additional electronic charge from the environment. The electrophilicity index encompasses both the propensity of the electrophile to acquire an additional electronic charge driven by μ^2^ and the resistance of the system to exchange electronic charge with the environment described by η, simultaneously [29]. In a way, a good electrophile, such as **9**, is characterized by a high value of μ (−4.98 eV) and a low value of η (2.84 eV). The structures of this study show the following order of reactivity: **9** > **10** > **4** > **1** > **8** > **3** > **2**. Finally, considering all these reactivity properties, it is appropriate to comment that the two new compounds, **9** and **10**, display a better PARP-1 inhibitory activity, and these properties support the previous results, highlighting that **9** is a good PARP-1 inhibitor candidate.

## 3. Materials and Methods

### 3.1. Molecular Dynamics Simulations of Four PARP-1 Inhibitors Related to Perezone Derivatives

The nature of ligand–receptor interaction, the binding affinity of the ligand towards the receptor, and the ligand stability in the active site of the receptor are basic requirements for rational drug design. For this purpose, the three-dimensional (3D) structure of PARP-1 was retrieved from the Protein Data Bank (PDB ID: 1UK0). The water molecules, co-crystalized ligands, and peptides were eliminated. The structures of perezone, perezone angelate, hydroxyperezone, and hydroxyperezone monoangelate were optimized using quantum chemistry. The starting structure of the complexes was obtained by docking studies employing Autodock 4.2 [30]. A GRID-based procedure was utilized to prepare the structural inputs and define all the binding sites. A rectangular lattice (70 × 70 × 70 Å) with points separated by 0.375 Å was centered on the active site of PARP-1. All docking simulations were conducted using the hybrid Lamarckian genetic algorithm with an initial population of 100 randomly placed individuals and a maximum of 1.0 × 10^7^ energy evaluations. All other parameters were maintained at their default settings. The lowest energy cluster for each protein–ligand was selected as the starting structure for MD simulations using the Nanoscale Molecular Dynamics (NAMD) Simulation 2.6 program (http://www.ks.uiuc.edu/Research/namd/ accessed on 17 October 2023) (Champaign-Urbana, IL, USA) [31], employing the CHARMM27 force field [32].

For the MD simulations of PARP-1 in absence of the ligands, hydrogen atoms were added to the initial coordinates of the protein using the psfgen program included in the Visual Molecular Dynamic (VMD) 1.8 program [33]. Afterwards, the system was solvated using TIP3P water molecules, and these structures were neutralized using X Na^+^. The equilibration protocol consisted of 1500 minimization steps followed by 30 ps of MD simulations at 10 K with fixed protein atoms. Subsequently, the entire system was minimized over 1500 steps (at 0 K), followed by gradual heating from 10 to 310 K using temperature reassignment during the initial 60 ps of the 100 ps equilibration molecular dynamics without restraints.

The snapshots, RMSD, and RMSF were obtained using Carma software (http://utopia.duth.gr/glykos/Carma.html accessed on 17 October 2023) (Glykos’ group, Alexandroupolis, Greece) based on Equations (1) and (2) [34].
(1)$$RMSF=\sqrt{\frac{1}{N}\sum_{t\;j=1}^{T}\left(x_{i}\left(t_{j}\right)-\tilde{x_{i}}\right)}$$
(2)$$RMSD=\sqrt{\frac{1}{N}\sum_{i}^{N}\delta_{i}^{2}}$$

The RMSD was analyzed to determine whether the protein had undergone conformational changes because this value reflects the distance between the pairs of the same atoms, represented by δ with time, Equation (1). Furthermore, RMSD values depicted the equilibration phases that determine the MD simulation quality. The RMSF indicated the displacement of each particle (Cα) to the structure, Equation (2). Therefore, the RMSF average was taken over time to provide a displacement for each particle (Cα per residue), whereas the RMSD average was calculated over the whole particle to provide time-specific values.

All computational work was performed using CUDA on an Intel Core i7-980x 3.33 GHz. Linux workstation with 12 Gb of RAM, (2×) NVIDIA GeForce GTX580 video cards, and a (1×) NVIDIA GeForce GTX680 video card. The parameters for perezone and related compounds to perform the MD simulations of the ligand–protein complex were obtained by employing Swiss Param Server [35].

### 3.2. Quantum Chemical Calculations

A conformational analysis at the molecular mechanic level, using Spartan06 (Version 06, Wavefunction, Inc., Irvine, CA, USA) [36] program, was performed. The Becke’s three-parameter hybrid density functional, B3LYP [37,38], of the DFT level of theory [39,40,41] with 6-311++G(d,p) method [42,43,44] using Gaussian 16 (Version 2016.03, Gaussian, Inc., 340 Quinnipiac St., Bldg. 40, Wallingford CT 06492) [45] software, and visualized by GaussView [46] application, was used to calculate geometry optimization and electronic properties: molecular electrostatic potential (MEP) [47], natural population analysis (NPA) [48], natural bond orbital (NBO) analysis [44] implemented in Gaussian 16 software, frontier molecular orbital [49], and DFT global chemical reactivity descriptors [50,51]. The optimization was carried out using the Berny analytical gradient optimization methods [52]. In all cases, default convergence criteria were used. Vibrational analysis showed that all the structures exhibit a minima on the potential energy surface (no imaginary frequencies). Also, a single-point energy calculation was performed.

### 3.3. Design of Compounds

According to MD simulations results to evaluate the interactions that drive the recognition of PARP-1 to compound **1** to **4**, a family of the perezone derivatives was designed. For this purpose, the sulfur substituent, Figure 5, was incorporated instead of the hydrogen atom on C6 of **1**. Perezone derivatives were submitted to docking studies with PARP-1, according to the previously mentioned procedure. The lowest energy cluster for each ligand was subjected to further free energy and binding geometry analyses, as was previously reported [53]. Conformations with the lowest free energy binding (ΔG) and the highest frequency were selected employing Autodock tools [30]. The images were created employing PyMol [54].

### 3.4. Synthesis of Compounds ***8***–***10***

#### 3.4.1. General

Starting materials phenylmethanethiol, 4-mercaptophenylacetic acid, and 5-fluoro-2-mercaptobenzyl alcohol were purchased from Sigma Aldrich Chemistry (St. Louis, MO, USA) and used as received. The solvents *n*-hexane, ethyl acetate, and absolute methanol were technical grade and were used without further purification from Materiales y Abastos Especializados S.A. de C.V. (Zapopan, Jalisco, Mexico). Perezone was isolated from the *Acourtia platyphilla* specimen and identified by their corresponding spectroscopical data correlated with the literature [55,56]. The reactions were monitored using thin-layer chromatography (TLC) performed on pre-coated Merck silica gel 60F254 aluminum sheets, and the visualization was achieved using a 254 nm UV lamp (UVLS-24, Upland, CA, USA), and purified with column chromatography using silica gel (0.063–0.200 mm, 70–230 mesh ASTM, acquired from Merck-Millipore, Germany). ^1^H and ^13^C (CDCl_3_) spectra were recorded using a Bruker Advance III spectrometer (San Francisco, CA, USA) at 300 MHz and 75 MHz, respectively. The multiplicities are reported as singlet (s), broad singlet (bs), doublet (d), triplet (t), double of double (dd), triplet of double (td), and multiplet (m). The corresponding δ-chemical shifts are given in ppm, employing TMS as internal reference. The EIMS (70 eV) were determined using a JEOL JMS-700 MStation mass spectrometer (JEOL, Tokyo, Japan). The HRMS-DART^+^ (19 eV) data were obtained using a JEOL AccuTOF JMS-T100LC (Direct Analysis in Real Time) spectrometer. The measurements were performed using the DART^+^ experiment with PEG (polyethylene glycol) 400 as an internal reference at 6000 resolutions and triplet helium as a carrier gas at 350 °C. In the first orifice, the temperature and voltage were 120 °C and 15 V, respectively, and the voltage in the second orifice was 5 V. Elemental composition was calculated within a mass range ±10 ppm from the measured mass. The IR spectra were recorded on a Bruker TENSOR 27 spectrophotometer (Bruker, Billerica, MA, USA).

#### 3.4.2. Synthesis of **9** and **10**

In an appropriate glass vessel, 1.0080 mmol (250 mg) of perezone, 0.98101 mmol (155 mg) of 5-fluoro-2-mercaptobenzyl alcohol or 1.0119 mmol (170 mg) of 4-mercaptophenylacetic acid, and 10 mL of absolute methanol were mixed, and then, the mixtures were treated under reflux by 8 h. The reaction progress was monitored using TLC (*n*-hexane/ethyl acetate, 95:5). The corresponding products were purified with column chromatography using the same mobile phase as the TLC and were obtained as red oils, Figure 1.

#### 3.4.3. Spectroscopic Characterization of **9** and **10**

2-((4-fluoro-2-(hydroxymethyl)phenyl)thio)-5-hydroxy-3-methyl-6-(6-methylhept-5-en-2-yl)cyclohexa-2.5-diene-1,4-dione (**9**), red oil was obtained with 45% isolated yield (184 mg); IR (cm^−1^): 3303 (OH), 2050 (C=C aromatic), 1724 (C=O), 1600 (C=C), 739 (C-S); ^1^H NMR (CDCl_3_) δ (ppm): 7.38–7.33 (dd, 1H, H21, *J* = 6 Hz), 7.28 (s, 2H, OH), 7.25–7.24 (d, 1H, H18, *J* = 3 Hz, J), 6.96–6.89 (td, 1H, H20, *J* = 3Hz), 5.28 (t, 1H, H12), 4.74 (s, 2H, H22), 2.95 (m, 1H, H8), 2.19 (s, 3H, H7), 1.80–1.64 (m, 4H, H10, H11), 1.64 (s, 3H, H14), 1.50 (s, 3H, H15), 1.09 (d, 3H, H9); ^13^C NMR (CDCl_3_) δ (ppm): 165.4 (C1), 162.1 (C4), 145.8 (C6), 145.7 (C19), 136.3 (C3), 136.2 (C21), 128.9 (C13), 128.8 (C17), 125.2 (C2), 124.8 (C12), 115.5 (C5), 115.3 (C16), 115.2 (C18), 115.0 (C20), 62.6 (C22), 34.0 (C10), 29.7 (C8), 26.6 (C11), 25.6 (C14), 24.8 (C9), 17.6 (C15), 14.0 (C7); EIMS (70 eV) *m*/*z* %ra [assignment]: 404 (48) M^+•^, 386 (12) [M-18]^+•^, 314 (149) [M-90]^+•^, 166 (27) [M-238]^+•^, 140 (100) [M-264]^+•^; HRMS-DART^+^ (19 eV): elemental composition of C_22_H_26_F_1_O_4_S_1_ [M + 1]^+^ for protonated molecule, exact value of 405.15313 Da, and precise value of 405.15358 Da, with an error of −1.12 ppm, and complementarily, the provided unsaturation data of 10.5.

2-(4-((hydroxy-2-methyl-5-(6-methylhept-5-en-2-yl)-3,6-dioxocyclohexa-1,4-dien-1-yl)thio)phenyl)acetic acid (**10**), red oil was obtained with 65% isolated yield (272 mg); IR (cm^−1^): 3372 (CO_2_H), 2110 (C=C aromatic), 1707 (C=O), 1637 (C=C), 738 (C-S); ^1^H NMR (CDCl_3_) δ (ppm): 13.83 (bs, 1H, CO_2_H), 7.22–7.19 (d, 2H, H18, H20, *J* = 6 Hz), 7.14–7.12 (d, 2H, H17, H21, *J* = 6 Hz), 7.06 (s, 1H, OH), 4.95 (t, 1H, H12), 3.54 (s, 2H, H22), 2.94 (m, 1H, H8), 2.02 (s, 3H, H7), 1.83–1.63 (m, 4H, H10, H11), 1.56 (s, 3H, H14), 1.44 (s, 3H, H15), 1.07 (d, 3H, H9); ^13^C NMR (CDCl_3_) δ (ppm): 182.5 (C1=O), 181.7 (C4=O), 176.7 (CO_2_H), 150.9 (C6), 146.6 (C3), 140.7 (C16), 133.2 (C19), 132.7 (C13), 131.4 (C5), 130.9 (C18, C20), 130.3 (C17, C21), 125.7 (C2), 124.4 (C12), 40.5 (C22), 34.1 (C10), 30.1 (C8), 26.7 (C11), 25.7 (C14), 18.3 (C9), 17.7 (C15), 14.3 (C7); EIMS (70 eV) *m*/*z* %ra [assignment]: 414 (89) M^+•^, 357 (12) [M-57]^+^, 332 (100) [M-82]^+•^, 166 (55) [M-248]^+•^; HRMS-DART^+^ (19 eV): elemental composition of C_23_H_27_O_5_S_1_ [M+1]^+^ for protonated molecule, exact value of 415.15669 Da, and precise value of 415.15792 Da, with an error of −2.96 ppm, and complementarily, the provided unsaturation data of 11.5.

### 3.5. PARP-1 Inhibition Assay

To corroborate the PARP-1 inhibition by **8**–**10**, PARP-1 activity was determined in their absence and presence, employing a colorimetric assay kit (BPS Bioscience Inc., San Diego, CA, USA, Catalog Number 80580), according to manufacturer’s instructions, employing Olaparib (BPS Bioscience Inc., San Diego, CA, USA, Catalog Number 27003) as a reference compound at crescent concentrations (0.00005, 0.0001, 0.0005, 0.001, 0.005, and 0.01 μM, n = 3). The selected compound was employed at final concentrations of 0.5, 1, 5, 10, 50, and 100 μM (n = 3). Absorbances were obtained at 450 nm using a UV/Vis spectrophotometer E max Precision microplate reader (Molecular Devices, San Jose, CA, USA).

### 3.6. Cytotoxic Evaluation on Cancer Cell Lines

To determine the inhibitory concentration 50 (IC_50_) of the compounds, the MTT assay (Creative Biolabs, Shirley, NY, USA, life technologies M6494) was performed on the A-549, MDA-MB-231, MCF-7, and Hela cell lines, which were cultured on 96-well microtiter plates (Glendale, AZ, USA, Corning 009017) with DMEM F-12 medium (Thermo Fisher Scientific, Waltham, MA, USA, gibco 11320-033) with 1% Antibiotic–Antimycotic (100-079) enriched with 10% FBS (fetal bovine serum, gibco 16141-079) and were incubated at 37 °C and 5% CO_2_ until reaching 90% confluency. Eight concentrations of the compounds **8**–**10** dissolved in culture medium were added in 1:2 dilutions, starting from 400 µM (400, 200, 100, 50, 25, 12.5, 6.25, and 3.125 µM) and keeping the remaining wells as controls (DMEM F-12 without FBS) and incubated for 48 h. At the end of this time, 20 µL of the MTT solution (5 mg/mL) was added to each well and incubated for 4 h in the dark at 37 °C and 5% CO_2_. The supernatant was decanted, and the resulting formazan was dissolved in isopropyl alcohol with 4 mM HCl solution and was read in a UV/Vis reader at 590 nm. The results obtained were plotted, and the IC_50_ was obtained through linear regression [13].

## 4. Conclusions

In the present study, in the first stage, the perezone (**1**), perezone angelate (**2**), hydroxyperezone (**3**), and hydroxyperezone monoangelate (**4**) were studied with molecular dynamic to verify their inhibitory activity on PARP-1 protein and analyze their potential as antineoplastic agents. In addition, the stability of protein–ligand complex structures during the simulation period of 50 ns was confirmed. Also, the compounds were subjected to theoretical studies: molecular electrostatic potential and charges that were obtained with the Density Functional Theory using the B3LYP hybrid method and with the set of base functions 6-311++G(d,p), to know the sites that could interact with the atoms of the amino acid residues of the PARP-1 protein. In terms of the obtained results from the molecular dynamics and quantum chemistry studies of compounds **1**–**4**, a set of eleven sulfur derivatives of perezone was proposed. Thus, the docking studies of the strategic compound set indicated that molecules **8**–**10** showed a higher ΔG values. Consequently, they were selected to be synthesized, characterized, evaluated in vitro, and analyzed at the quantum level (molecular orbitals and reactivity properties). In vitro testing with PARP-1 showed that the IC_50_ of PARP-1 of compounds **9** (0.317 µM) was lower compared to that of compounds **1**–**4**, **8**, and **10**. In addition, **9** exhibited inhibitory activities against MDA-MB-231 and MCF-7 cells.

According to the studies of the reactivity properties and the frontier orbitals of systems **1**–**4** and **8**–**10**, it was observed that compounds **8**–**10** were more reactive, and this improved their affinity for PARP-1, and therefore, the IC_50_ showed by the sulfur derivates of perezone was lower. With the development of this study, compounds **8**–**10** were determined to be prospective antineoplastic candidates.

## Data Availability

The data reported in this study are available upon request to nicovain@yahoo.com.mx.

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
