# Peer review of "Design of Two New Sulfur Derivatives of Perezone: In Silico Study Simulation Targeting PARP-1 and In Vitro Study Validation Using Cancer Cell Lines"

_ijms, 2024, doi:10.3390/ijms25020868_

Round 1

Reviewer 1 Report

Comments and Suggestions for Authors

Suggestions for improving Manuscript (MS):

1. The title is still absurd and please revise the title to:

Design of two new sulfur derivatives of perezone: In silico Simulation by targeting PARP-1 and in vitro study validation on cancer cell lines

2. Why in silico only Molecular Dynamics? what about Molecular Docking? why don't authors do or present Molecular Docking Simulation?

3. Why are there no control drugs such as Doxorubicin or Mitoxhantrone in the cell lines test?

4. A concentration gradient of sample compounds for cell line testing must be provided.

5. Please change the subtitle "3.6 Determination of IC50 in cancer cell lines" to "3.6 In Vitro Study on cancer cell lines" and state how you calculate the IC50 as the statistical analysis methods. 

Reviewer 2 Report

Comments and Suggestions for Authors

Nicolás-Vázquez et al. have described the

design and preparation of two new sulfur derivatives of perezone,  studying the structure, in silico and in vitro biological activity

evaluation. The synthesis of the investigated compounds is not novel in this manuscript (Nat. Prod. Commun. 2008, 3, 1465-1468).

 The quality of this paper is generally good although only two compounds are described. I would like to recommend the acceptance of this manuscript after responding to the comments below:

 For better insight for the reader I suggest to including a general chemical formula of perezone derivatives, where R indicates the location of and type of sulfur-containing fragment.

 The docking calculations performed are described in detail for compounds 8-10 only. It would be useful to comment briefly on the data for the remaining compounds.

 The synthesis of compound 8, as well as its spectral characteristics in the sections 3.4.2 Synthesis and 3.4.3 Spectroscopic characterization should not be given as they have already been described.

Tables 1, 2 and 3 should be placed as supplementary information. The information from the tables is present in the text

Reviewer 3 Report

Comments and Suggestions for Authors

Authors presented valuable data on new inhibitors of poly-ADP-ribose polymerase, important target of some types of cancer. Sesquiterpene quinone Perezone was shown to induce apoptosis in several cancer cell lines at low concentrations.

Based on computational study, authors proposed a variety of alkylthio and arylthio substituents for incorporation in the structure of perezone derivatives.

Molecular docking studies of eleven sulfur perezone derivatives with PARP-1 protein were performed.

Phenylmethanethiol, 5-fluoro-2-mercaptobenzyl alcohol, and 4-mercaptophenylacetic acid derivatives were selected for detailed study, compounds were prepared through Michael addition approach.

Three sulfur perezone derivatives were studied by spectroscopy methods, their interaction with amino acid residues of PARP-1 was estimated. For these compounds PARP-1 inhibition assay was accomplished, Olaparib was taken as reference drug. 4-Fluoro-2-(hydroxymethyl)phenyl)thione derivative demonstrated both the lowest in vitro IC50 towards PARP-1 (0.317 μM) and the lowest band gap energy (2.554 eV).

The manuscript can be published after minor revision. Some comments:

1Line 207. Do you mean free binding energies from -11.78 to -5.71 kcal/mol ?

2Line 270. The chapter 2.5 is mainly devoted to spectral characteristics, proof of structure, it is incorrect to indicate only synthesis in the title.

3Line 498. The synthesis of compound 8 was reported previously, for this reason the sentence in discussion should be organized in more appropriate style (synthetic method included…)

4Figure 10 was not cited in the text. HOMO and LUMO are presented, so “orbitals” would be more correct in the title.

5According to data of Table 6, conclusion in the line 590 sounds not very convincing.

Comments on the Quality of English Language

single-plural
